# Safety Data in Patients with Autoimmune Diseases during Treatment with High Doses of Vitamin D3 According to the “Coimbra Protocol”

**DOI:** 10.3390/nu14081575

**Published:** 2022-04-10

**Authors:** Ulrich Amon, Raul Yaguboglu, Madeleine Ennis, Michael F. Holick, Julian Amon

**Affiliations:** 1International Centre for Skin Diseases DermAllegra, Coimbra Protocol Certified Center, Am Markgrafenpark 6, 91224 Pommelsbrunn-Hohenstadt, Germany; yaguboglu@dermallegra.de (R.Y.); julian.amon@aol.com (J.A.); 2The Wellcome-Wolfson Institute for Experimental Medicine, Queens University of Belfast, Belfast BT7 1NN, UK; m.ennis@qub.ac.uk; 3Endocrinology, Diabetes, Nutrition & Weight Management, Department of Medicine, Boston University School of Medicine, Boston, MA 02118, USA; mfholick@bu.edu

**Keywords:** autoimmune diseases, Coimbra protocol, high-dose therapy, multiple sclerosis, parathyroid hormone (PTH), vitamin D receptor (VDR), vitamin D, vitiligo, safety, single nucleotide polymorphisms (SNPs)

## Abstract

Background: In 2013, the group of Cicero Coimbra, Brazil, reported the clinical efficacy of high doses of vitamin D3 in patients suffering from autoimmune skin disorders (“Coimbra protocol”, CP). However, hypercalcemia and the subsequent impaired renal function may be major concerns raised against this protocol. Methods: We report for the first time for a broad spectrum of autoimmune diseases in 319 patients (mean age (±SD) 43.3 ± 14.6 years, 65.5% female, 34.5% male) safety data for high doses of orally applied vitamin D3 (treatment period: up to 3.5 years) accompanied by a strict low-calcium diet and regular daily fluid intake of at least 2.5 L. Results: Mean vitamin D3 dose was 35,291 ± 21,791 IU per day. The measurement of more than 6100 single relevant laboratory parameters showed all mean values (±SD) within the normal range for total serum calcium (2.4 ± 0.1 mmol/L), serum creatinine (0.8 ± 0.2 mg/dL), serum creatinine associated estimated GFR (92.5 ± 17.3 mL/min), serum cystatin C (0.88 ± 0.19 mg/L), serum TSH (1.8 ± 1 mIU/L), and for 24 h urinary calcium secretion (6.9 ± 3.3 mmol/24 h). We found a very weak relationship between the dosage of oral vitamin D3 and the subsequent calcium levels, both in serum and in urinary excretion over 24 h, respectively. Conclusions: Our data show the reliable safety of the CP in autoimmune patients under appropriate supervision by experienced physicians.

## 1. Introduction

The worldwide increase in patients with autoimmune diseases during the last decades [1] has led to a significant challenge to our health systems, both therapeutically and economically [2,3]. The etiology of autoimmune diseases is multifactorial, involving a combination of genetic and environmental factors. Numerous different mechanisms, including exposure to environmental pollution and toxins, the complex mechanisms of lifestyle factors (e.g., “Western diet”, smoking and alcohol consumption, psychosocial stress), infections, and intestinal dysbiosis, are attributed to initiate complex pathogenetic cascades leading to systemic or organ-specific autoimmune conditions [4,5].

Independent of a broad spectrum of therapeutic strategies for the many different autoimmune diseases that we can offer our patients today, an integrative approach to treatment should be our goal. Although it is questionable whether the pathogenetic complexity of all autoimmune diseases can be focused on one central or specific factor, there is accumulating evidence for an important regulatory role of the biologically active hormone/Vitamin D (1,25-dihydroxyvitamin D3;1,25(OH)2D3) in this context [6].

Vitamin D has a complex role in the immune system, regulating both innate and adaptive immunity and resulting in the inhibition of inflammation and the enhancement of defense mechanisms [7,8]. Mediated by the vitamin D receptor (VDR), 1,25(OH)2D3 can influence immune function as well as the differentiation and growth of many cell types in addition to its well described central role in bone metabolism [7,9,10].

We and others have demonstrated an association of serum levels of 25(OH)D and numerous chronic diseases, including inflammatory skin diseases (e.g., psoriasis and atopic dermatitis), autoimmune reactions (e.g., rheumatoid arthritis, multiple sclerosis, and vitiligo), inflammatory conditions as well as cardiovascular diseases, diabetes mellitus, metabolic syndrome, certain malignant tumors (e.g., breast, colon, and prostate), diseases of the central nervous system brain (e.g., schizophrenia, Alzheimer’s disease, and depression), and infections (upper respiratory tract, tuberculosis, and COVID-19) [9,11,12,13]. Vitamin D status is also considered to be of high importance for longevity [14,15].

In 2013, the group of Cicero Coimbra in São Paulo, Brazil, reported the clinical efficacy of high doses of vitamin D3 together with dietary instructions in patients suffering from autoimmune skin disorders (psoriasis and vitiligo), the so-called “Coimbra protocol” (CP) [16,17]. In short, in the study, 25 patients received vitamin D3 35,000 IU once daily for six months in association with a low-calcium diet and sufficient hydration. After treatment, 25(OH)D3 levels significantly increased from 14.9 ± 7.4 to 106.3 ± 31.9 ng/mL and from 18.4 ± 8.9 to 132.5 ± 37 ng/mL in patients with psoriasis and vitiligo, respectively. Parathyroid hormone (PTH) levels inversely decreased significantly from 57.8 ± 16.7 to 28.9 ± 8.2 pg/mL and from 55.3 ± 25.0 to 25.4 ± 10.7 pg/mL in patients with psoriasis and vitiligo, respectively. The clinical status of the patients as measured by the PASI Score for psoriasis and the degree of repigmentation for vitiligo improved significantly [16].

Coimbra and co-workers, after two decades of clinical experience, state that the therapeutic approach of the protocol relies on doses of vitamin D that range from 40,000 IU to 300,000 IU per day [17,18]. The conventional starting dose in multiple sclerosis, for example, is approximately 1.000 IU vitamin D3 per kg body weight [19].

In Germany, this therapeutic protocol for autoimmune patients has been used since 2016 [20]. Underlying the CP is the hypothesis of the non-hereditary, but acquired form of vitamin D resistance and insufficient biological activity of 1,25(OH)2D3, which both may be overcome by high doses of vitamin D3, compensating the resistance. Lemke et al. have recently elegantly elaborated the background of an acquired vitamin D resistance as a possible cause of autoimmune diseases and could confirm this hypothesis of the malfunction of the vitamin D metabolism underlying the efficacy of high-dose vitamin D CP [19].

According to their proof of concept, single nucleotide polymorphisms (SNPs) within the genes of the vitamin D system (e.g., in activating enzymes, serum transport, and VDR) cause individual vitamin D resistance and reduced vitamin D responsiveness. PTH serum levels serve as valuable biomarker for vitamin D3 dosing and its effect on altering calcium metabolism [19].

Modern technologies have revealed hundreds of thousands of common genetic SNPs variations, including VDR in autoimmune patients [20,21,22]. To date—apart from the high-dose vitamin D protocol—no other causal treatment for correcting a blockade of the VDR is known.

Since 1,25(OH)2D3 centrally regulates calcium/phosphate homeostasis, hypercalcemia may be a major concern raised against the CP. However, as intensively discussed in the paper of Lemke et al. [19], vitamin D resistance appears to confer an intrinsic protection against hypercalcemia. Following the precautionary advice for the patients, such as avoidance of dairy products and a minimum fluid intake of 2.5 L/day as well as continuous monitoring of blood parameters and kidney function, a framework for the protocol has been established to reduce the risk of hypercalciuria and hypercalcemia for the patients [19].

Safety data over a longer time scale during the treatment with the CP for a broad spectrum of autoimmune diseases have not yet been published. Hence, in this paper, we report a retrospective analysis of almost 300 patients monitored with respect to their treatment according to the CP as well as an analysis of gene polymorphisms (SNPs) of the vitamin D metabolism in a subgroup of patients.

## 2. Patients and Methods

The aim of this study was to document the safety parameters during a high-dose treatment with vitamin D3. In our center, we have been working holistically, including regular measurements of 25(OH)D as well as complementary vitamin D3 supplementation for ca. 10 years. After an intense specific educational program, we started using the CP for patients with autoimmune diseases in March 2018.

The high-dose vitamin D protocol was performed according to the very extensive personal experience of Coimbra and co-workers as described [19], whilst also respecting the Declaration of Helsinki (chapter 37: “Unproven interventions in clinical practice”) [23]: “In the treatment of an individual patient, where proven interventions … have been ineffective, the physician, after seeking expert advice, with informed consent from the patient or a legally authorised representative, may use an unproven intervention if in the physician’s judgement it offers hope of saving life, re-establishing health or alleviating suffering…”.

Based on information available on the Internet [20], patients specifically sought out our center to be treated with the high-dose vitamin D protocol. All patients were informed orally by the physician in a lengthy session lasting at least 60 min and by a detailed patient information sheet, given a few weeks before the start of the study. All patients provided a signed informed consent. The first patient of the study was included at the end of March 2018. Before obtaining an appointment, all patients had to fill out a questionnaire to rule out any contraindications for the CP, e.g., impaired renal function, disturbed calcium metabolism, low daily fluid intake, refusal to restrict dietary calcium intake by eliminating milk and milk products and other foods with a high calcium content from their diet. The next step required a recent complex laboratory investigation as baseline before starting the CP. These parameters included, among others, complete blood count, ferritin, albumin, renal and liver function tests, cystatin C, 25(OH)D, parathormone, electrolytes, including serum calcium and phosphate, vitamin B12, selenium, thyroid gland hormones, TSH, and renal calcium secretion over 24 h. Baseline bone densitometry was also performed.

Provided there were no contraindications with respect to the personal history or baseline laboratory results, the CP was usually started by prescribing oral vitamin D3 in the following daily initial dose: 1000 IU/kg body weight for MS; 300–1000 IU/kg body weight for the majority of other autoimmune diseases, such as rheumatoid arthritis, psoriatic arthritis, connective tissue diseases, plaque psoriasis, inflammatory bowel diseases; and 150–300 IU/kg body weight for autoimmune inflammation of the thyroid gland [19]. The latter dosage was usually the starting dose in children for all diagnoses.

In addition to advice about living a healthy lifestyle (e.g., healthy diet and physical activity if possible), patients were given the following detailed information and instructions [16]:To drink at least 2.5 liters of fluid per day (with a calcium level less than 200 mg/L). In the case of fever, profuse sweating (for whatever reason) or gastrointestinal infection, this amount must be increased or, in the event of illness, supplemented with infusions if necessary. This volume must be maintained even during long-haul flights, longer car trips, bus and train journeys, etc.To avoid foods rich in calcium (maximum intake ca. 500 mg per day). A low-calcium diet, which avoids milk and dairy products (some butter is allowed), food and dietary supplements with added calcium, is necessary in order to prevent excessive absorption of calcium from the intestine that could cause hypercalciuria and hypercalcemia as an automatic consequence of a high-dose vitamin D therapy and to protect renal function. In addition, patients were told not to eat peanuts, walnuts, almonds, chestnuts, cashews, nut-granola bars, pistachios, hazelnuts, etc., and dried fruits with seeds as well as sesame paste (tahini), hummus, baba ghanoush, sardines and anchovies. They were also recommended to avoid the consumption of green smoothies (with a high concentration of dark green vegetables, such as kale and spinach). Calcium supplementation was not allowed.To regularly attend visits for laboratory investigations.

In order to avoid osteopenia or osteoporosis following strict calcium restrictions, frequent and regular exercise (jogging and walking) or where physically applicable daily use of a vibration plate was strongly recommended as prophylaxis.

In addition to dietary controls, it was equally important for the patients to manage stress levels. Since chronic stress or a permanent stressful situation, fear, anger, and depression can have a very negative effect on the therapy, patients were educated to reduce and prevent stress by different methods (e.g., meditation, yoga, qigong, tai chi, and psychotherapy).

Since magnesium is an essential cofactor for 25-hydroxylase (CYP2R1) and 1a-hydroxylase (CYP27B1) to generate 1,25(OH)2D3 [24], and hypomagnesemia is probably the most underdiagnosed electrolyte deficiency in current medical practice [25], it is essential to ensure that magnesium (glycinate, malate, citrate, etc.) is added (elemental magnesium 200–300 mg QID).

There are potential synergies between vitamin A and vitamin D in the modulation of tissue-specific immune responses, leading to the prevention and/or treatment of inflammation and autoimmunity [26,27,28,29]. To maintain immunological homeostasis, a balance between these two immune regulatory factors should be considered [30,31]. We therefore also prescribe ca. 9000 IU retinyl palmitate per day (=ca. 20.000 µg per week), since the tolerable dose for vitamin A intake to prevent teratogenicity in females (UL value) is 10,000 IU retinol per day [29,32].

The final supplement that we prescribe is vitamin K2. Vitamin K2 (menaquinone) has been described as a protective factor for bones (cofactor for mineralization in synergy with vitamin D and vitamin A), the circulatory system and endothelium (cofactor for demineralization, also in synergy with vitamin D and vitamin A) [33,34,35]. In addition, antioxidative effects have been described [36]. As recently shown, MS patients have much lower blood levels of vitamin K2 than healthy controls [37]. Since high doses of vitamin D may increase the risk of artery calcification following the elevation of serum calcium [38], we add vitamin K2 to the CP in daily concentrations between 100 µg and 800 µg depending on the starting dose of vitamin D3 and the levels of serum calcium and urinary calcium excretion.

Further dietary supplements depend on many different aspects, such as type and disease activity, blood analysis, degree of inflammation, oxidative and nitrosative stress, results from gut microbiome analysis and many others.

Regular laboratory monitoring after starting the protocol include, but are not limited to PTH, total and ionized serum calcium, serum phosphate, parameters for renal function (blood urea nitrogen, creatinine, estimated glomerular filtration rate, and cystatin C), serum albumin, ferritin, TSH, and 24 h renal calcium excretion. Depending on baseline levels, bone densitometry as well as bone metabolizing parameters (such as bone-specific alkaline phosphatase, P1NP or osteocalcin, and urine for serum crosslinks) should be monitored individually. Additionally, ultrasound examination of the kidneys once a year is very valuable.

The vast majority of our patients requesting CP treatment live more than 100 km away from our center. Thus, their local general practitioners are usually responsible for the measurement of laboratory markers. They are regularly informed by letter about the individual treatment after the first appointment. Since the majority of our patients have to pay for all laboratory controls themselves (due to health insurance regulations in Germany), as minimal standard, the following laboratory values are always required: total serum calcium, creatinine with estimated glomerular filtration rate, cystatin C, TSH, and PTH. To avoid an interference in measuring PTH, patients are told to discontinue all micronutrients containing biotin (= vitamin H = vitamin B7) about 7 days before taking the blood sample. Normally, the first determinations after starting of the protocol occurred at week 6 to 8 and approximately every three months during the treatment. Depending on the starting dose of vitamin D3, urinary calcium secretion is measured 12 weeks after beginning the treatment and every four-to-six months during the treatment. Serum concentrations of 25(OH)D are not regularly required since the values increase in all patients. In contrast to PTH, calcium and renal parameters, the degree of the increase in 25(OH)D serum levels during the treatment was not used for evaluation of both efficacy and side effects of the CP [16,19].

Blood and urine samples for analysis were collected in the local GP laboratory responsible for each patient in Germany, Austria and Switzerland.

All laboratory results are sent by email to our center. Depending on the parameters, feedback is given by the authors (U.A., R.Y.) concerning the further dosage of vitamin D3 as well as advice about possible implications for diet, fluid intake and the date for the next measurements for blood/urine samples.

We performed a retrospective analysis of routine laboratory samples of 319 patients with different autoimmune diseases treated according to the CP between March 2018 and June 2021, focusing on the routine blood and urine samples collected after starting CP.

In a subgroup of 130 patients, we were able to analyze the frequency of different SNPs of the vitamin D metabolism [39] in a specialized laboratory (Biovis Diagnostik MVZ GmbH, Limburg-Offheim, Germany) in order to clarify whether and to what extent vitamin D-related gene variants might influence the CP. We investigated the following genes: 25-hydroxylase (CYP2R1, rs2060793) [40], 1α-hydroxylase (CYP27B1, rs703842) [41], 24-hydroxylase (CYP24A1, rs2296241) [42], vitamin D binding protein (VDBP, GC, rs7041, rs1155563, rs4588) [43,44,45], and VDR (BsmI rs1544410, rs731236 TaqI, rs2228570 FokI) [46,47].

In the statistical analyses, we had to deal with the issue that the time intervals between laboratory controls inevitably varied between patients. For this purpose, the observations of laboratory parameters were grouped according to the number of days since the start of the protocol (Initial measurements: 0–30 days, first half year: 31–180 days, first year: 181–365 days and greater than one year: >365 days). In each of these time intervals, all measurements of laboratory controls are included. Hence, multiple measurements from the same patient are part of the sample if the values were checked more than once in the corresponding period. For the statistical analyses, such instance of multiple per-patient measurements in a given period were averaged. Differences in parameters between the initial measurements and subsequent periods were then analyzed on a per-patient basis using Wilcoxon signed-rank tests. Differences between independent samples were analyzed using Wilcoxon rank sum tests. All analyses were performed with the open-source software R.

## 3. Results

### 3.1. Population

#### 3.1.1. Sex, Age, Diagnosis

The majority of patients of our 319 patients (age range 11–85 years, mean age 43.3 ± 14.6 standard deviation (SD) years, 209 female (65.5%): 44.5 ± 14.3 years, 110 male (34.5%): 40.9 ± 15.25 years) suffered from vitiligo or multiple sclerosis with 36% and 23%, respectively (Figure 1). In 230 cases, we found a second and in 66 cases a third autoimmune disease (data not shown). Patients suffering from more than three autoimmune diseases have not been separately documented in this paper.

#### 3.1.2. Vitamin D and Parathormone

A total of 58.9% of all patients had been taking a vitamin D3 supplement (regardless of the dose) at least within 3 months before they began the CP treatment (“pre-treated”), which had an impact on the baseline serum levels of 25(OH)D and PTH. Mean serum levels (±SD) for 25(OH)D and PTH were 40.3 ± 25.8 ng/mL and 42.3 ± 19.9 pg/mL in pre-treated patients and 26.4 ± 20.7 ng/mL and 49.5 ± 23.9 pg/mL in patients without vitamin D pre-treatment, respectively. The differences were highly significant: *p* < 0.001 for 25(OH)D and *p* = 0.003 for PTH (Wilcoxon rank sum test). Further on, because of the high dosage of vitamin D3, a differentiation of vitamin D supplementation before baseline was no longer necessary.

Vitamin D3 was taken orally according to dose recommendations of CP [19] at 1000 IU/kg body weight for MS and 300–1000 IU/kg body weight for the majority of other autoimmune diseases. The maximum dose in our population was 150,000 IU per day (Table 1), mean dosage for all patients was 35,291 ± 21,791 (SD) IU per day, whereas mean dosage for patients with MS was significantly higher than for patients without MS (52,955 ± 25,791 IU per day vs. 29,683 ± 16,861 IU per day, *p* < 0.0001, Table 1).

Although we did not explicitly encourage GPs involved in laboratory controls to measure 25(OH)D serum levels, we had the chance to evaluate 440 values of 25(OH)D in 186 different patients with a mean of 141.4 ± 75.6 (SD) ng/mL (data not shown). The maximum value of 806 ng/mL 25(OH)D in a single female patient with MS was not paralleled by abnormal results in renal function or serum and urinary calcium levels.

### 3.2. Safety Data during the Treatment

#### 3.2.1. Laboratory Values

In 319 patients, more than 6100 single relevant laboratory parameters were determined and evaluated. Overall, mean values (±SD) of measurements for total serum calcium (2.4 ± 0.1 mmol/L), serum creatinine (0.8 ± 0.2 mg/dL), serum creatinine associated estimated GFR (92.5 ± 17.3 mL/min), serum cystatin C (0.9 ± 0.2 mg/L), serum TSH (1.8 ± 1 mIU/L) as well as for 24 h urinary calcium excretion (6.9 ± 3.3 mmol/24 h) remained within the normal range (Figure 2).

When median baseline laboratory values were compared with values during treatment with the CP, the parameters for renal function and calcium metabolism did not show significant changes (Figure 3).

The onset of clinical symptoms suggestive of hypercalcemia (increased thirst, constipation, nausea, and vomiting) was not reported at any time.

To obtain all blood test parameters from different GP practices and different laboratories at baseline and during treatment resulted in many difficulties, since often single values were missing or incorrect parameters had been determined, although all patients had been provided with written instructions many weeks before their first appointment. Among all parameters necessary for evaluation of any risks at the beginning of the treatment, the determination of the 24 h urinary calcium excretion appeared to be most complicated for the majority of peripheral GP practices where the follow up parameters had been taken. However, in a subgroup of patients, we obtained all serum and urine measurements at baseline as well as the follow-up values, which were determined in the same laboratory.

When we then compared the parameters for single patients at baseline with their average follow-up values, we could not detect renal impairment or any clinically relevant changes in the serum calcium and 24 h urinary excretion (Figure 4).

Among more than 6100 single laboratory values, the increase in renal calcium excretion was the most sensitive parameter that was used to either decrease or stop vitamin D3 administration. Statistical calculations revealed consistently weak Spearman correlations between the dosage of vitamin D3 and serum calcium levels and renal calcium excretion (Table 1). Although some of the observed correlations are significantly different from zero (mostly due to the large sample size), all of them are smaller than 0.25 in absolute value. This further confirms the safety of high doses of orally applied vitamin D3, as increases in dosages are—under appropriate doctoral supervision—only moderately correlated with the subsequent serum and urinary calcium measurements.

However, with respect to all single measurements in 319 patients in over 3.5 years, we temporarily stopped vitamin D3 only in 27 situations, when calcium excretion exceeded more than 10 mmol/24 h (normal values: 2.50–8 mmol Calcium/24 h). In almost all cases, the dietary calcium intake was reviewed with the patient with emphasis on reducing calcium intake to less than 500 mg daily and they were encouraged to increase fluid intake. As a result, the patients were restarted on the CP four-to-eight weeks later without any further disruptions in their treatment. It is notable that a 24 h urine calcium as a marker is helpful for monitoring the possible side effects during CP, but finally not ideal, since the parameter could be under- or overestimated if the collection was less or more than 24 h. For further studies, the regular determination of the urine calcium–urine creatinine ratio might be a more sensitive measure to evaluate the need of protocol interruption.

In some cases, we also included patients with slight calciuria (8–15 mmol Calcium/24 h), which had been treated before by other CP centers. Under further instructions, none of them had to stop the CP (data not shown).

#### 3.2.2. Discontinuation of Treatment

In 16 cases (13 f, 3 m, mean age 39.1 years), the CP treatment was stopped due to different reasons (e.g., non-compliance, no clinical effect, increase in symptoms, food supplements too expensive, diet too complex, daily fluid intake too stressful, pregnancy, *familial hypocalciuric hypercalcemia*, and significant increase in parameters of bone metabolism).

### 3.3. Parathormone Levels and Vitamin D3 Dosage during Treatment

PTH served as a good surrogate for monitoring the effect of high doses of vitamin D3 on calcium metabolism. As shown (Figure 5 and Table 2), there was a significant decrease (*p* < 0.001) in PTH levels from baseline with for patients with MS or non-MS patients.

While PTH levels plateau, the vitamin D3 dose significantly drops over time to maintain this effect, which might underline the epigenetic influence of the CP (Figure 6 and Table 3).

This effect was much more prominent for patients with MS, where higher vitamin D3 dosages were used (baseline vs. > 365 days *p* = 0.001; non-MS-patients: baseline vs. > 365 days *p* = 0.024), reflecting also a more prominent drop in PTH levels over time in comparison to baseline (Figure 5). Clinically, this stands for a mean reduction in a daily vitamin D3 dose of 16,318.5 IU within the observation period for patients with MS in comparison to baseline dosage.

### 3.4. Single Nucleotide Polymorphisms

A subgroup of 130 patients agreed to be investigated using routine gene analysis for single nucleotide polymorphisms within the vitamin D metabolism. A total of 73.9% showed more than four mutations in nine different genes (Figure 7). Seven patients had mutations in all genes investigated. The most frequent mutations were found in the genes encoding the enzymes 25-Hydroxylase (CYP2R1) and 1alpha-Hydroxylase (CYP27B1) with 75% of all patients (CYP2R1: 32% homozygote and 43% heterozygote; CYP27B1: 23% homozygote and 52% heterozygote, data not shown).

In genes encoding the vitamin D binding protein (VDBP, GC), we found homozygote mutations in 23% (SNP rs7041), 8% (SNP rs1155563), and 10% (SNP rs4588) of all patients (data not shown).

In different regions of the VDR, the detection of SNPs was less frequent, showing 53%, 63% and 68% combined homozygote and heterozygote mutations for BsmI, TaqI and FokI, respectively.

Due to the small numbers of subgroups, we could not find significant correlations between the quantity of mutations, on the one hand, and the initial PTH level, on the other hand, in patients having been pre-treated with vitamin D3 before baseline (Pearson’s R −0.133, *p* = 0.517, *n* = 26) and patients without vitamin D3 supplements before baseline (Pearson’s r −0.219, *p* = 0.221, *n* = 33), respectively.

## 4. Discussion

Although the pathophysiological role and the therapeutic implications of vitamin D in autoimmunity are still under debate [6], the pleiotropic non-skeletal functions of vitamin D have been generally recognized for reducing the risk of complex non-communicable diseases (NCDs), including cardiovascular diseases, diabetes mellitus, depression, dementia, cancer, allergies, asthma as well as chronic infections [48,49,50].

While for most NCDs the putative pathophysiologic role of vitamin D as being an association or a cause–effect relationship is still unclear, recent evidence, using Mendelian randomization studies focusing on the genetic variants of vitamin D metabolism in order to reduce confounders, has confirmed an increased risk at least for MS in patients with vitamin D deficiency [51].

Therefore, MS might be the optimal “model” to study both the effects and safety parameters of the high-dose vitamin D treatment as offered with the CP [19]. To date, the treatment protocol of Cicero Coimbra in Brazil for a variety of autoimmune disorders, e.g., MS, psoriasis, and vitiligo, has only been published as pilot study [16], and randomized clinical trials are missing.

Patients with autoimmune diseases, especially MS, often look for holistic and alternative approaches in addition to symptomatic, immune suppressive and disease modifying treatments [52]. Nowadays, very extensive information is available in the World Wide Web [53,54,55]. There are many websites and personal case reports, which reflect the efficacy of CP as seen in social media by the patients [53,54,55].

Since the increasing interest in treatment with vitamin D and even ultra-high doses [7,56] results in the possible consequences of self-administering highly concentrated food supplements, we here primarily focus on the safety aspects of the CP.

In the pilot study of Coimbra’s group published in 2013 [16], assessing the effect of the prolonged administration of high daily doses of vitamin D on the clinical course of vitiligo and psoriasis, nine patients with psoriasis and sixteen patients with vitiligo received vitamin D3 35,000 IU once daily for six months in association with a consequent low-calcium diet (avoiding dairy products and calcium-enriched foods, such as oat, rice or soya “milk”) and reliable hydration (minimum 2.5 L daily). In their study, serum creatinine and calcium (total and ionized) did not change and urinary calcium excretion increased within the normal range [16].

To our knowledge, we in this paper demonstrate for the first time for a broad spectrum of autoimmune diseases in over 300 patients with a treatment period of more than 3.5 years that high doses of orally applied vitamin D3 up to 1000 IU per kg bodyweight are safe in terms of calcium metabolism and renal function, when strict recommendations for diet and fluid intake are followed, up to a treatment period of 3.5 years.

With respect to our population, the mean daily dose of 35,000 IU Vitamin D3 is far higher than usual recommendations [57] and should be a contribution to the debate on the disparity of conclusions on what an “optimal” serum concentration of 25(OH)D is and how much supplementation is required to achieve a sufficient clinical response without long-term side effects.

For many years, we have been following clinically the recommendation of 25(OH)D ranging between 40 and 60 ng/mL being considered to be “optimal” (preferred range) for healthy people [9,58,59,60]. However, for autoimmune and inflammatory diseases, we have little knowledge about the variations of vitamin D metabolism and recognition, such as VDR polymorphisms, vitamin D binding protein polymorphisms, extrarenal 1 alpha hydroxylase activity, and micro RNAs [61].

Instead of solely focusing on 25(OH)D serum levels, different groups have recently elegantly worked out that, due to epigenetic and genetic differences, the individual immune responsiveness to vitamin D3 is rather complex [19,57,62,63]. Among other factors, this depends on the individual ability to convert vitamin D to its active metabolite 1,25(OH)2D and the interaction with VDR and the response elements [62,63,64]. Recently, it was shown that there is a dissociation between the calcemic and non-calcemic biologic actions of vitamin D3, especially on functions involved in immune activity [57].

Against this background, the suppression of PTH should be rather favored as a proxy for optimal vitamin D status as well as vitamin D3 treatment [65].

As expected [19], PTH levels dropped over time during CP treatment depending on the dose of vitamin D3 in our population. This confirms the preliminary results of Coimbra’s group [16]: during a six month treatment period with a fixed daily dose of 35,000 IU vitamin D3 PTH levels significantly decreased from 57.8 ± 16.7 to 28.9 ± 8.2 pg/mL and from 55.3 ± 25.0 to 25.4 ± 10.7 pg/mL in patients with psoriasis and vitiligo, respectively.

Shirvani and co-workers [57] described a plateau in PTH levels in thirty randomized healthy adults at 16 weeks for a vitamin D3 dose of 4000 and 10,000 IU per day, but not for the group that received 600 IU per day, respectively, without changes in serum calcium. Interestingly, they observed a dose-dependent 25(OH)D alteration in broad gene expression with 162, 320 and 1289 genes up- or down-regulated in white blood cells, respectively [57].

However, due to higher oral vitamin D3 dosages in our patients suffering from MS, in our protocol, PTH plateaued at 15 pg/mL at approximately 6 months after start of treatment, which was different from non-MS autoimmune patients (Figure 5).

In comparison to healthy persons in autoimmunity genetic or epigenetic alterations of vitamin D metabolism differs both in quantity and quality of SNPs within genes of the vitamin D system (e.g., in activating enzymes, serum transport, and VDR) responsible for vitamin D status alterations, causing vitamin D resistance and reduced vitamin D responsiveness [7,19]. The influence of vitamin D3 high dose supplementation on genome-wide expression in autoimmune patients remains unexplored.

As already stated in the pilot study of Finamor et al. [16], the daily requirements of vitamin D3 for patients with autoimmune disorders should be individually adapted to the profile of genetic polymorphisms of vitamin D metabolism.

The association between VDR, SNPs, and MS risk, for example, has been reported by many groups, whereas other vitamin D-related genes (including CYP2R1, CYP27B1, CYP24A1) have been less investigated [39].

In our population, the most frequent mutations were found in the genes encoding the enzymes 25-Hydroxylase (CYP2R1) and 1alpha-Hydroxylase (CYP27B1) with 75% of all patients, while SNPs in the regions of the VDR (BsmI, TaqI and FokI) were detected less frequently.

As previously shown, the genetic polymorphism of CYP27B1 associated with autoimmunity [66,67] causes a relative resistance to vitamin D requiring a higher level of circulating 25(OH)D3 to achieve biologically active 1,25(OH)2D3, resulting in normalized immune functions [16]. In order to achieve a physiologic rate of product formation in polymorphic enzyme variants, a higher *K*_m_ (decreased affinity for substrate) and/or a lower *V*_max_ require supra-physiologic concentrations of the substrate [16].

In patients with a combination of polymorphisms within different sections of vitamin D metabolism, this effect is potentiated. Supra-physiologic doses as applied according to CP may compensate for this genetic-related status of relative vitamin D resistance establishing tolerance to auto-antigens and may with respect to our safety data also increase tolerability in patients with autoimmune disorders [16]. This might explain the far lower PTH plateau in our patients in comparison with healthy subjects studied by Shirvani and co-workers [57].

According to Lemke et al. and the hypothesis of Cicero Coimbra of an acquired vitamin D resistance in autoimmune diseases [19], PTH concentrations could be used as a hallmark for individual adaption of oral vitamin D3 dosages. For an optimal physiological response of 1,25(OH)2D3, a low PTH plateau should reached and maintained within the lower third of the reference range [19]. With respect to our own experiences during the last four years, the degree of inflammation in autoimmune processes seem to influence the need to further decrease the PTH level by higher daily doses of vitamin D3.

A seven-year experience of McCullough et al. with oral vitamin D3 up to 50,000 IU per day did not reveal a linear or exponential relationship between vitamin D and calcium blood levels [68]. They did not observe cases of vitamin D3 induced hypercalcemia or any adverse events attributable to vitamin D3 supplementation in any patient.

Historically, many reports were published during the last century describing the successful use of vitamin D3, for example, in treating psoriasis, asthma, or rheumatoid arthritis with daily doses ranging from 60,000 to 300,000 IU [68]. Due to serious concerns following complications from vitamin-D-induced hypercalcemia after the prolonged daily intake of these supra-physiologic daily doses, vitamin D was then labelled as toxic [68].

With our current detailed knowledge about vitamin D metabolism, central cofactors (e.g., magnesium [69]), the influence of SNPs according to (epi)genetic studies [39], and worldwide experience of several thousands of patients treated with the CP from 2012 onwards with daily doses up to 340,000 IU [70], we are able to develop an individualized vitamin D3 treatment for autoimmune patients by the careful planning and determination of reliable mechanisms for regular laboratory controls. Based on our findings, hypercalcemia does not appear to be a first line risk of high-dose vitamin D3 therapy.

However, since hypervitaminosis D leads to increased calcium absorption via the upregulation of intestinal VDR, a strict calcium-reduced diet is mandatory to protect patients from hypercalcemia. At baseline, patients must be informed in great detail about this central aspect. Restrictions in milk, dairy products and calcium-enriched food stuff have contributed to minimize the calciotropic effects of high daily doses of vitamin D3 in the current study, which confirms the data of the pilot study of Coimbra’s group [16]. We strongly recommend that CP is always used in the hands of qualified and experienced physicians and strongly advise against the use of CP by patients themselves based on Internet information.

Among all the chronic inflammatory skin diseases we previously studied (Amon U et al., 2018), the average 25(OH)D serum level was lowest in patients with plaque psoriasis (psoriasis vulgaris), only in patients with severe acute or chronic recurrent skin infections on the skin did we find significantly lower levels. Since the 1930s [71], successful oral substitution of vitamin D in psoriasis patients has been demonstrated in different studies [72]. Since for psoriasis, an increased incidence of other inflammatory comorbidities (e.g., psoriatic arthritis, arteriosclerosis, diabetes mellitus, obesity, non-alcoholic steatohepatitis, inflammatory bowel diseases, and depression) has been described [73], for which—independent of psoriasis—a modulating role of vitamin D has also been discussed in recent publications [74], we strongly recommend that physicians consider using the CP for patients suffering from psoriasis [16].

Perez and co-workers observed an improvement of 88% after oral 1,25(OH)2D3 treatment in 85 patients with psoriasis, with approximately 27% of patients healing completely, 36% having a moderate and 25% showing a slight improvement without causing hypercalcemia [75]. High oral doses of vitamin D3 (35,000 IU once daily) for six months led to a significant improvement of Psoriasis Area and Severity Index (PASI) score in nine patients with psoriasis without significant side effects [75]. Earlier work has also demonstrated a drastic improvement following the oral administration of vitamin D2 in severe cases [71] and following the use of 1α-hydroxyvitamin D3 and 1,25(OH)D3 [76,77].

In our analysis, vitiligo was the dominant diagnosis. Vitiligo is a polygenic autoimmune disease, characterized by localized or generalized depigmentation of the skin as a result of melanocyte destruction by immunologic dysfunction [78]. Two recent meta-analyses identified a significant positive relationship between lower 25(OH)D serum levels and the incidence of vitiligo [79,80]. In our recent study, among 113 patients with vitiligo, 41.2% had a 25(OH)D serum level below 20 ng/mL at baseline before treatment [11].

Zhang et al. performed a meta-analysis with 17 studies on VDR gene polymorphisms (BsmI, ApaI, TaqI, and FokI) in vitiligo, resulting in an increased susceptibility risk of vitiligo only for the Apal polymorphism of the VDR; BsmI, TaqI, and FokI loci had no obvious correlation [80]. In our study, FokI (rs2228570) was most often mutated in comparison to BsmI and TaqI.

When Finamor et al. treated 16 vitiligo patients with daily doses of 35,000 IE vitamin D3 over a period of six months, no repigmentation of the affected areas was observed in only five patients, but the others did develop repigmentation up to 75% without further dermatological treatment [16]. In our center, the combination of CP with topical treatment (calcineurin inhibitors) and/or 311 nm narrow band UVB or 308 nm excimer laser regularly leads to repigmentation in all patients, of course to a varying extent, which will be reported elsewhere. With respect to the data presented here, the vast majority of our vitiligo patients had a daily dose of vitamin D3 not higher than 40,000 IE. For non-inflammatory autoimmune skin diseases (such as vitiligo or alopecia areata), in general, a lower vitamin D3 dose appears to be sufficient for a good clinical response (drop of PTH levels in patients with vitiligo: baseline vs. >365 days *p* < 0.001, non-vitiligo patients: baseline vs. >365 days *p* < 0.001), as has been suggested elsewhere [16,19].

Despite our evidence that high-dose vitamin D3 application according to CP is generally well tolerated without signs for long term toxicity, this therapeutic approach of vitamin D3 supplementation should be embedded in a holistic treatment package. Against this background, fascinating new findings from gut microbiome research demonstrated variations in the vitamin D receptor also influence the large functional network of gut microbiota [81].

Finally, further possible significant factors influencing vitamin D sensitivity and metabolism as well as disease activity of autoimmune processes, such as gender differences [82] and the degree of chronic psychoneuroimmunologic impact [83], should be elaborated in further studies, when the clinical efficacy of CP is evaluated in detail. It will be also particularly important to differentiate the clinical effects between the different autoimmune diseases treated by the regimen.

## 5. Conclusions

In summary, to our knowledge, our work provides the first long-term documentation of selected critical laboratory parameters during the application of the CP using a high-dose oral vitamin D3 in a broad spectrum of different autoimmune diseases, demonstrating that this procedure is well tolerated with respect to renal function and calcium metabolism. In terms of individualized treatment, we suggest to further use serum levels of PTH as biomarker for an individual response to vitamin D3, the individual ability to convert vitamin D to the active metabolite, the 1,25(OH)2D’s interaction with its receptor and the response elements and finally the differential supplementation with vitamin D3. In further studies, possible differences of the clinical outcome of CP treatment should be investigated.

## Figures and Tables

**Figure 1 nutrients-14-01575-f001:**
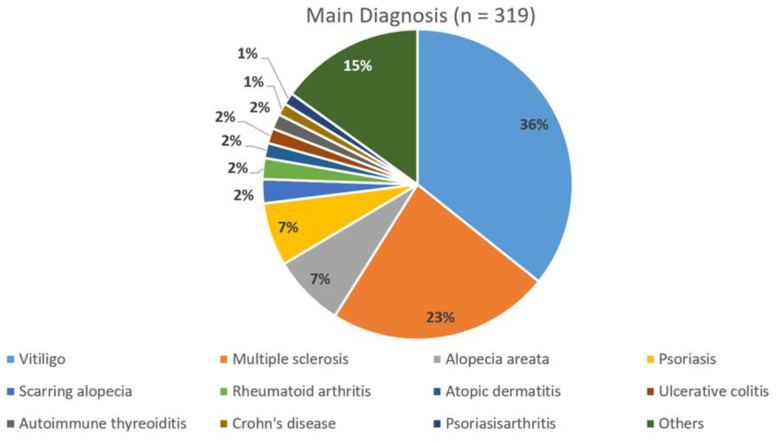
Distribution of primary (main) diagnosis (*n* = 319).

**Figure 2 nutrients-14-01575-f002:**
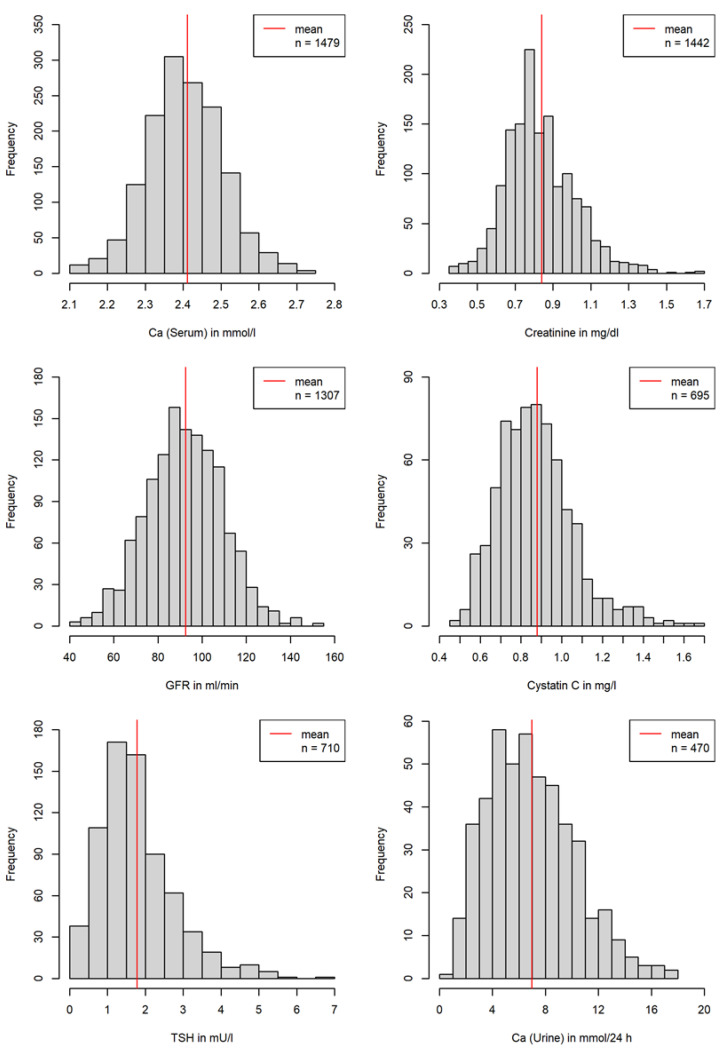
Serum parameters and 24 h urinary calcium secretion over all measurements. Normal values (may vary depending on laboratory methods): serum calcium 2.2–2.6 mmol/L, serum creatinine levels 0.5–1.3 mg/dL, serum creatinine associated estimated GFR 90–120 mL/min, serum cystatin C 0.5–0.96 mg/L, serum TSH 0.27–4.2 mIU/L, and renal calcium excretion within 24 h 2.50–8 mmol/24 h.

**Figure 3 nutrients-14-01575-f003:**
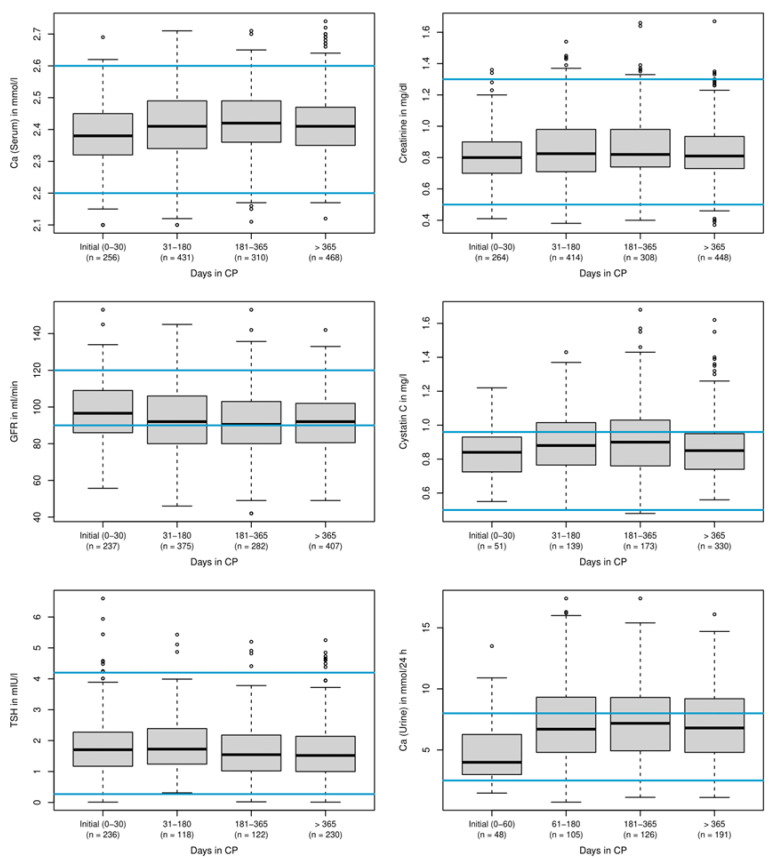
Changes of laboratory parameters during treatment. Detailed median values: serum calcium levels (mmol/L) initial (0–30): 2.38/31–180: 2.41/181–365 2.42/>365: 2.41; serum creatinine levels (mg/dL) initial (0–30): 0.80/31–180: 0.83/181–365: 0.82/>365: 0.81; serum creatinine associated estimated GFR (ml/min) initial (0–30): 96.60/31–180: 92.00/181–365: 90.50/>365: 92.00; cystatin C levels (mg/L) initial (0–30): 0.84/31–180: 0.88/181–365: 0.90/>365: 0.85; TSH levels (mIU/L) initial (0–30) 1.71/31–180: 1.73/181–365: 1.55/>365: 1.52; renal calcium secretion within 24 h (mmol/24 h) initial (0–60): 3.99/31–180: 6.70/181–365: 7.18/>365: 6.80. Blue lines: upper and lower normal values. Individual points outside the whiskers indicate outliers more than 1.5 times the interquartile range away from the upper quartile. See also legend of Figure 2.

**Figure 4 nutrients-14-01575-f004:**
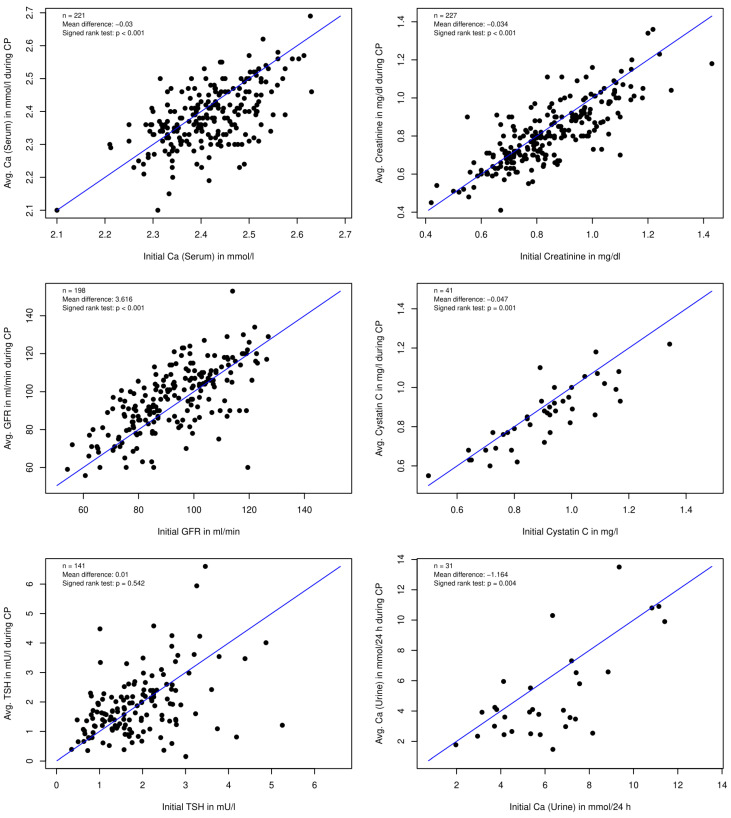
Comparison of individual baseline values of blood and urine parameters with individual average follow-up values.

**Figure 5 nutrients-14-01575-f005:**
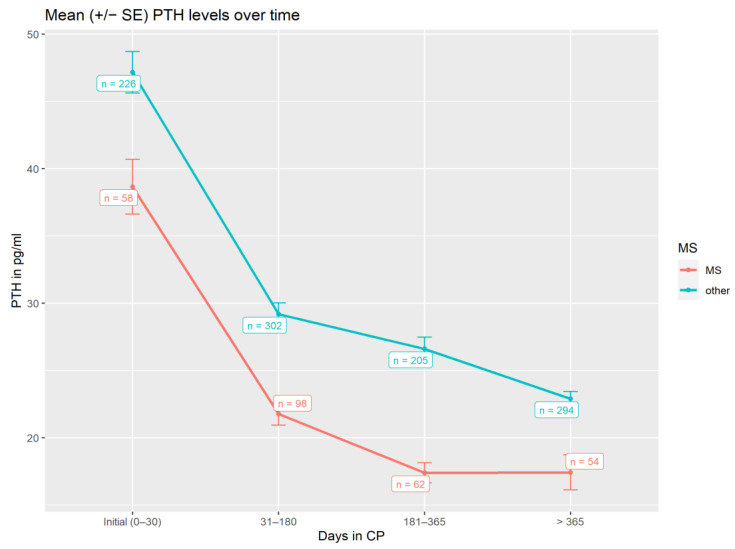
Serum PTH levels (pg/mL) over time during CP treatment, using all laboratory measurements available in the corresponding time frame.

**Figure 6 nutrients-14-01575-f006:**
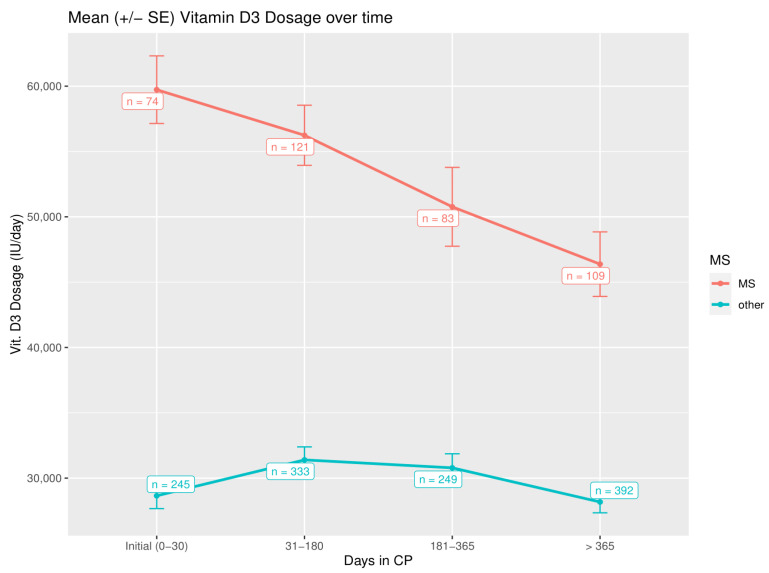
Vitamin D3 dosage (IU/day) over the time of the CP treatment, using all dosages determined after a laboratory investigation (*n* = 1606 in total).

**Figure 7 nutrients-14-01575-f007:**
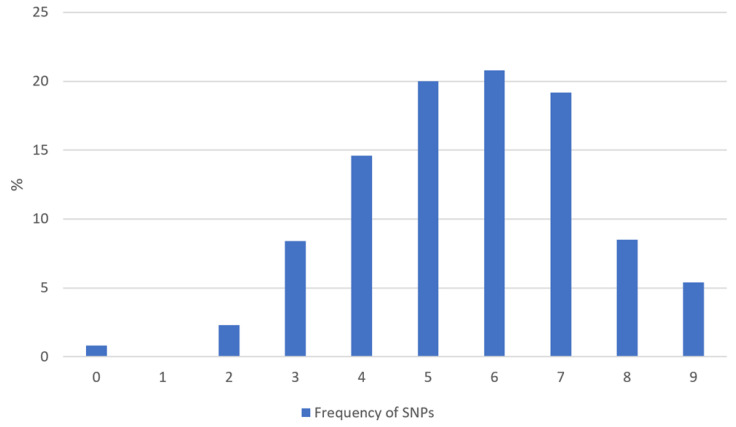
Frequency of detected homozygote or heterozygote single nucleotide polymorphisms in 9 different genes regarding vitamin D metabolism (*n* = 130).

**Table 1 nutrients-14-01575-t001:** Dosage of oral vitamin D3 in the present population with respect to all individual determinations of prospective vitamin D3 dose after having received the laboratory results.

	All Patients (*n* = 319)	MS Patients (*n* = 74)	Non-MS Patients (*n* = 245)
Number of laboratory investigations	1606	387	1219
Minimum/Maximum of daily vitamin D3 dosage (IU per day)	0/150,000	0/150,000	0/100,000
Mean dosage of daily vitamin D3 (IU per day ±SD)	35,291 ± 21,791	52,955 ± 25,791	29,683 ± 16,861
Spearman correlation dosage vitamin D3 and serum calcium	0.103, *p* < 0.001 *n* = 1204	0.151, *p* = 0.010 *n* = 291	0.050, *p* = 0.128 *n* = 913
Spearman correlation dosage vitamin D3 and urinary calcium excretion	0.162, *p* = 0.001 *n* = 433	0.058, *p* = 0.456 *n* = 166	0.233, *p* < 0.001 *n* = 267

In total, the 319 patients were subjected to 1606 laboratory investigations. The current dosage of oral vitamin D3 was either maintained or adapted after each of these investigations. The correlations presented in this table represent the Spearman correlations between the currently agreed upon dosage of oral vitamin D3 and the serum/urinary calcium levels measured at the subsequent laboratory follow-up, broken down by MS and non-MS patients. As the table shows, although some of the observed correlations are significantly different from zero (mostly due to the large sample size), all of them are moderate in terms of their absolute numerical value. In our sample, we thus only find a very weak relationship between the dosage of oral vitamin D3 and the subsequent calcium levels, both in serum and in urinary excretion.

**Table 2 nutrients-14-01575-t002:** Patient-level statistical analysis of serum PTH levels (pg/mL) over time.

	Mean Difference of Initial PTH and Average PTH 31–180 Days in CP	Mean Difference of Initial Dose and Average Dose 181–365 Days in CP	Mean Difference of Initial Dose and Average Dose >365 Days in CP
MS patients	−16.1 pg/mL PTH (*p* < 0.001)	−19.3 pg/mL PTH (*p* < 0.001)	−25.3 pg/mL PTH (*p* < 0.001)
Non-MS patients	−18.4 pg/mL PTH (*p* < 0.001)	−22.0 pg/mL PTH (*p* < 0.001)	−28.3 pg/mL PTH (*p* < 0.001)

While Figure 5 displays the PTH measurements in all laboratory investigations within a given time frame after protocol initiation, Table 2 presents the statistical results on a per-patient basis to account for the repeated measurement structure. For every MS and non-MS patient, all PTH measurements within a given time frame (e.g., 31–180 days in CP) are averaged. We then report the mean difference of these average PTH levels to the initial level across patients as well as the *p*-values resulting from a Wilcoxon signed-rank test.

**Table 3 nutrients-14-01575-t003:** Patient-level statistical analysis of vitamin D3 dosages (IU/day) over time.

	Mean Difference of Initial Dose and Average Dose 31–180 Days in CP	Mean Difference of Initial Dose and Average Dose 181–365 Days in CP	Mean Difference of Initial Dose and Average Dose >365 Days in CP
MS patients	−2881.4 IU (*p* = 0.039)	−9520.9 IU (*p* = 0.008)	−16,318.5 IU (*p* = 0.001)
Non-MS patients	−442.9 IU (*p* = 0.864)	+163.2 IU (*p* = 0.640)	−2736.8 IU (*p* = 0.024)

While Figure 6 displays the dosages determined after all laboratory investigations within a given time frame after protocol initiation, Table 3 presents the statistical results on a per-patient basis to account for the repeated measurement structure. For every MS and non-MS patient, all dosages within a given time frame (e.g., 31–180 days in CP) are averaged. We then report the mean difference of these average dosages to the initial dosage across patients as well as the *p*-values resulting from a Wilcoxon signed-rank test.

## Data Availability

To share and archive our results, datasets will be provided in anonymized form as links for download.

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
