# Peer review of "Safety Data in Patients with Autoimmune Diseases during Treatment with High Doses of Vitamin D3 According to the “Coimbra Protocol”"

_nutrients, 2022, doi:10.3390/nu14081575_

Round 1

Reviewer 1 Report

Reviewer’s opinion on MS Nutrients-1647823

Generally: this is an interesting study and a well written MS on vitamin D supplementation by Coimbra protocol in the treatment of autoimmune diseases. This is a treatment/ drug safety study.

In details: English is good.

Question is relevant, methods seem to be adequate, study design is well done. Number of patients is big, and length of observation time is long, resulting in strong evidence on this field.

In the introduction, the authors might mention that both cardiovascular disease and vitamin D sensitivity show a gender difference, with interactions observed between vitamin D deficiency and gender playing a role in gender differences in cardiovascular risk (PMID: 34360792).

Methods: Helsinki’s declaration is not necessary to cite in detail – (lines 112-118), only that part supporting your creed.

In the methods, the authors mention that patients have been involved in the treatment of chronic stress, - as chronic stress, fear, anger, and depression may negatively affect the therapy they use. According to literature data, to what extent does chronic stress affect skin symptoms? To what extent can stress treatment reduce skin symptoms?

There is a big standard deviation in patients’ age (11-85 ys) – water spaces are quite different. How do you correct the doses upon the age/ water spaces (adjust to creatinine or other way)? You should write in details the dose estimation used.

Table I., first line – strange data about sampling frequency of vitamin D – explain it in the table legend and check the data! Eg. MS: 110 pts/ 387 measurements/ 3,5 ys – if I understood correctly (cc. once a year)?

Figure 6. – non MS data on the Figure and in the figure legend text are different – homogenize these!

I suggest using homozygote instead of homocygote.

You mention that effectivity of vitamin D treatment should be estimated by PTH levels. You should give the optimal range in the discussion, you recommend (if this is disease-specific, mention it also).

Author Response

The authors are very greatful for the numerous very helpful comments. Please find a point-by-point discussion in the following paragraphs:

Point 1: In the introduction, the authors might mention that both cardiovascular disease and vitamin D sensitivity show a gender difference, with interactions observed between vitamin D deficiency and gender playing a role in gender differences in cardiovascular risk (PMID: 34360792).

Response: Instead of adding this important aspect to the introduction we mentioned the possible influence of gender difference in vitamin D sensitivity at the end of the discussion to point out its need for further investigations.

Changes: Text lines #607-610, ref. [86].

Point 2: Methods: Helsinki’s declaration is not necessary to cite in detail – (lines 112-118), only that part supporting your creed.

Response: We have shortened this citation.

Changes: Text lines #116-116 was deleted.

Point 3: In the methods, the authors mention that patients have been involved in the treatment of chronic stress, - as chronic stress, fear, anger, and depression may negatively affect the therapy they use. According to literature data, to what extent does chronic stress affect skin symptoms? To what extent can stress treatment reduce skin symptoms?

Response: These important questions touch on a very large scientific complex about decades of research. Since the present data focus on safety aspects using high therapeutic doses of vitamin D3 authors would suggest to discus the possible psychoneurologic impact in the context of clinical efficacy of the Coimbra protocol. A separate paper is currently in preparation. However, with respect to this comment we have changed the final paragraph of the discussion and added a scientific review on this topic.

Changes: Text lines #607-610, ref. [87].

Point 4: There is a big standard deviation in patients’ age (11-85 ys) – water spaces are quite different. How do you correct the doses upon the age/ water spaces (adjust to creatinine or other way)? You should write in details the dose estimation used.

Response: Authors are not quite sure what the reviewer means by "water spaces". Obviously, renal disease can alter the renal elimination of drugs and can lead to sub- or supratherapeutic drug concentration. It is also well known that creatine clearance rates go down with age. As outlined in our paper, we strictly ruled out any contraindications for the high dose vitamin D3 treatment at baseline, particularly impaired renal function or disturbed calcium metabolism. This was true for all age groups. Even if at baseline in older age groups parameters of renal function showed mild impairment, close monitoring of blood and urine parameters allowed us to individually adapt the daily dosage of vitamin D3 avoiding further decrease of kidney function. With special respect to children in our population, we clarified in the manuscript, that we usually used a startig dose of 150 - 300 IU vitamin D3 per kg body weight in this age group.

Changes: Text line #140.

Point 5: Table I., first line – strange data about sampling frequency of vitamin D – explain it in the table legend and check the data! Eg. MS: 110 pts/ 387 measurements/ 3,5 ys – if I understood correctly (cc. once a year)?

Response: The information of each laboratory investigation is used to make a decision on the oral dosage of vitamin D3 going forward, i.e. whether it should remain the same or be adapted. Therefore, for the 74 MS patients, for instance, there were in total 387 laboratory investigations and corresponding reviews of the current vitamin D3 dosage in all of their time in the protocol. A statement on how many measurements occurred per year cannot be ascertained from this table, as the patients vary in how long they have been in the protocol and how many measurements have been performed. The table thus only gives information on the aggregate numbers.

Changes: Table 1 (text lines #276-289) was adapted to make this more clear. In particular, we included the number of patients in each group in the table header, changed the description of the first line for better clarity and extended the table legend to better explain the numbers presented in the table.

Point 6: Figure 6. – non MS data on the Figure and in the figure legend text are different – homogenize these!

Response: The lack of homogenity between the information presented in the figure and the figure legend text was a result of two different levels of analysis: the figure contained the mean (and SE) of all dosages determined in a given time frame (following a laboratory investigation), so that the displayed n values add up to the 1,606 (387 MS and 1,219 non-MS) in table 1. The figure legend text had the goal of statistically analyzing whether the decreases of dosage over time were significant. It would be statistically incorrect, however, to simply perform t-tests or Wilcoxon tests comparing the mean dosages over all measurements as displayed in figure 6 because this neglects the repeated measurement structure. Instead, we averaged all dosages in a given time frame on a per-patient basis and then report the mean differences across patients along with a p-value resulting from a signed rank test.

Changes: We acknowledge that reporting these numbers in the figure legend text without any further explanation was confusing, so we created table 3 (text lines #394-399), which now explains these analyses in detail and reports the results separately from figure 6. Moreover, as the same procedure was also applied to the PTH levels over time and reported in the legend text of figure 5, we also separated the different levels of analysis for figure 5. Accordingly, table 2 (text lines #376-381) now contains the patient-level results for serum PTH levels with a fitting explanation, while the legend text of figure 5 (text lines #369-375) now only mentions that all laboratory measurements in a given time frame were used. Similarly, the former legend of figure 6 was also shortened (text lines #387-393).

Point 7: I suggest using homozygote instead of homocygote.

Changes: All instances of the words “homocygote” and “heterocygote” have been replaced correspondingly (text lines #412-418, #426).

Point 8: You mention that effectivity of vitamin D treatment should be estimated by PTH levels. You should give the optimal range in the discussion, you recommend (if this is disease-specific, mention it also).

Response: As Lemke et al. [ref. #19 in our paper] have elegantly worked out elevated PTH concentrations in autoimmune patients could be used as a hallmark of acquired vitamin D resistance, even if 25(OH)D3 serum levels being in the ideal range. As it is well known, PTH constitutes a direct feedback mechanism within the vitamin D system. If 25(OH)D3 serum levels are high, PTH should be usually low and vice versa. In patients with autoimmune diseases this negative feedback loop is disturbed. Based on these observations, Prof. Coimbra proposed the hypothesis of a vitamin D resistance. For an optimal physiological response of of 1,25(OH)2D3, PTH should be lowered into the lower third of the reference range.

Because reference ranges and measurement units for PTH vary among different laboratories, it could be stated that a 25(OH)D3 serum value of >40 ng/ml should be associated with a PTH value in the middle of the lower third of its laboratory-specific reference range. The reduction of PTH concentrations into the middle of the lower third of its laboratory-specific reference range is judged as an indicator for overcoming vitamin D resistance. For example, if a laboratory’s reference range for PTH is 15-65 ng/ml, the middle of the lower third of that range would be 23.3 ng/ml. With respect to our own experiences during the last four years the degree of inflam-mation in autoimmune processes seem to influence the need to further decrease the PTH level by higher daily doses of vitamin D3.

Changes: We have clarified this aspect and added a new paragraph in the discussion (text lines #528-534)

Reviewer 2 Report

The work confirms that vitamin D is not easy to overdose and that the system of metabolic self-regulation is effective. The presented results are very important from the therapeutic point of view, especially with such a wide range of vitamin D activity and its participation in the etiopathogenesis of a number of diseases. It also undoubtedly sheds new light on the safety of high doses of vitamin D and provides topics for reflection whether, for example, the most frequently found hypercalcemia does not result from the omission of certain elements of patient preparation, such as the supply of calcium, vitamin K2 or magnesium or there is some specific metabolic profile that provides resistance to high doses of vitamin D.

It should be emphasized well-thought-out design of the study that takes into account all issues that may be important for the condition of patients, also resulting from the dietary restrictions imposed in the study, such as: protection against calcium deficiency and the consequences for bone formation by recommended physical exercises, or vitamin K2 supplementation; Mg supplementation, vitamin A supplementation, bone density measurement, etc..

In the present study, results on patients with various autoimmune diseases are developed and discussed together, although in the group of patients with MS they are discussed separately. In subsequent works, comparative studies of responses to high-dose vitamin D supplementation should be planned, with a distribution to specific diseases. In a combined analysis of autoimmune diseases, a number of important correlations may be overlooked, such as the relationship between disease pathophysiology and serum calcidiol levels or specific mutations in the genes associated with vitamin D.

Author Response

The work confirms that vitamin D is not easy to overdose and that the system of metabolic self-regulation is effective. The presented results are very important from the therapeutic point of view, especially with such a wide range of vitamin D activity and its participation in the etiopathogenesis of a number of diseases. It also undoubtedly sheds new light on the safety of high doses of vitamin D and provides topics for reflection whether, for example, the most frequently found hypercalcemia does not result from the omission of certain elements of patient preparation, such as the supply of calcium, vitamin K2 or magnesium or there is some specific metabolic profile that provides resistance to high doses of vitamin D.

It should be emphasized well-thought-out design of the study that takes into account all issues that may be important for the condition of patients, also resulting from the dietary restrictions imposed in the study, such as: protection against calcium deficiency and the consequences for bone formation by recommended physical exercises, or vitamin K2 supplementation; Mg supplementation, vitamin A supplementation, bone density measurement, etc.

The authors are very greatful for the kind comments. Please find a discussion in the following paragraph:

Point 1: In the present study, results on patients with various autoimmune diseases are developed and discussed together, although in the group of patients with MS they are discussed separately. In subsequent works, comparative studies of responses to high-dose vitamin D supplementation should be planned, with a distribution to specific diseases. In a combined analysis of autoimmune diseases, a number of important correlations may be overlooked, such as the relationship between disease pathophysiology and serum calcidiol levels or specific mutations in the genes associated with vitamin D.

Response: It will be particularly important for our further studies to differentiate clinical effects between the different autoimmune diseases treated by the regimen. A paper focusing on clinical efficacy of the Coimbra protocol is currently under preparation.

Changes: Text lines #612-614.